# Molecular Abnormalities in BTBR Mice and Their Relevance to Schizophrenia and Autism Spectrum Disorders: An Overview of Transcriptomic and Proteomic Studies

**DOI:** 10.3390/biomedicines11020289

**Published:** 2023-01-20

**Authors:** Polina Kisaretova, Anton Tsybko, Natalia Bondar, Vasiliy Reshetnikov

**Affiliations:** 1Institute of Cytology and Genetics, Siberian Branch of Russian Academy of Sciences, Prospekt Akad. Lavrentyeva 10, Novosibirsk 630090, Russia; 2Department of Natural Sciences, Novosibirsk State University, Pirogova Street 2, Novosibirsk 630090, Russia; 3Department of Biotechnology, Sirius University of Science and Technology, 1 Olympic Avenue, Sochi 354340, Russia

**Keywords:** BTBR, transcriptome, proteome, cortex, hippocampus, ASD, schizophrenia

## Abstract

Animal models of psychopathologies are of exceptional interest for neurobiologists because these models allow us to clarify molecular mechanisms underlying the pathologies. One such model is the inbred BTBR strain of mice, which is characterized by behavioral, neuroanatomical, and physiological hallmarks of schizophrenia (SCZ) and autism spectrum disorders (ASDs). Despite the active use of BTBR mice as a model object, the understanding of the molecular features of this strain that cause the observed behavioral phenotype remains insufficient. Here, we analyzed recently published data from independent transcriptomic and proteomic studies on hippocampal and corticostriatal samples from BTBR mice to search for the most consistent aberrations in gene or protein expression. Next, we compared reproducible molecular signatures of BTBR mice with data on postmortem samples from ASD and SCZ patients. Taken together, these data helped us to elucidate brain-region-specific molecular abnormalities in BTBR mice as well as their relevance to the anomalies seen in ASDs or SCZ in humans.

## 1. Introduction

Autism spectrum disorders (ASDs) are a group of lifelong neurodevelopmental conditions characterized by dysfunction of communication and of social interaction as well as restricted repetitive behavior and interests [1]. Despite high variation of ASD prevalence across the world, the general tendency has been a dramatic increase of this prevalence in recent decades [2]. The global burden of ASDs is substantial and has continued to grow over the past three decades as well [3].

Schizophrenia (SCZ) is associated with such symptoms (present for at least 6 months) as delusions, hallucinations, disorganized speech and behavior, and dysfunctions of social and occupational domains [4]. Although ASDs and SCZ are recognized as separate disorders with divergent clinical profiles, growing evidence suggests that SCZ is a neurodevelopmental disorder as well [5,6,7,8,9]. Moreover, the convergence of ASDs and SCZ at certain levels supports the idea that their phenotypes may overlap [10,11]. Specific genetic risk alleles shared among intellectual disability, ASD, SCZ, attention deficit/hyperactivity disorder, and bipolar disorder have led investigators to propose the model of neurodevelopment continuum, in which these diseases represent a range of outcomes that flow from aberrant brain development [12]. Despite some studies showing that the burden of DNA copy number variants (CNVs) positively correlates with the severity of childhood neurodevelopmental disorders [13,14,15] (in line with the continuum hypothesis), this model may be an oversimplification of the diagnostic conundrum. At the same time, a close connection between ASDs and SCZ is likely. SCZ alone is three-to-six times more common in people with ASDs than in controls, as demonstrated in two meta-analyses [16,17]. A recent systematic review showed that ASDs and SCZ significantly overlap in behavior, perception, cognition, some biomarkers, and genetic risk factors [18]. These findings indicate that SCZ and autism are on the same continuum.

Given that estimated heritability is 50%, genetic factors are the main contributors to ASD etiology [19]. Nonetheless, autism has substantial phenotypic heterogeneity, which arises from multiple genetic sources and many of them overlap [20]. From the perspective of the complex genetic architecture of autism, Iakoucheva et al. have proposed the omnigenic model of ASDs, which posits inseparability of effects of genes which have an impact on the trait directly and gene modifiers which act indirectly through gene regulation [21]. On the other hand, on the basis of 28 meta-analyses, Qiu et al. recently identified 12 significant single-nucleotide polymorphisms (SNPs) in nine candidate genes [22], most of which fit the definition of “core” genes. Thus, the probability of identification of at least several key genes for the ASD phenotype is still not negligible. In this context, transcriptomic and proteomic approaches provide insights into molecular characteristics to help bridge the gap between genes and functions. There are several transcriptomic articles about postmortem brain samples from ASD subjects and much fewer proteomic ones [23]. The number of comparative transcriptomic studies on brain samples from both ASD and SCZ patients is relatively small [24,25,26,27], and there is only a single systematic multi-omics study on ASD and SCZ [28]. 

Both ASD and SCZ are polygenic disorders, which means that a great variety of genes engaged in different processes are involved in phenotype development. In ASD alterations in expression of glutamate decarboxylases GAD65/GAD67 and GABAA and GABAB receptor subunits are considered the most reproducible. For SCZ, the most consistent genes are trophic factor *NRG1* and post-synaptic tyrosine kinase receptor *ERBB4*. Genetic studies identified genes affected in both disorders (*DISC1*, *NRXN1*, *NLGN3-4*, *SHANK3*, and *CNTNAP2*) [29].

One of the most widely used animal models of idiopathic ASDs is the BTBR T^+^ Itpr3tf/J (BTBR) inbred mouse strain. The phenotype of BTBR mice is relevant to major diagnostic symptoms of ASDs, including reduced social interactions and stereotyped behavior [30,31,32]. BTBR mice share neuroanatomical features with a subgroup of ASD patients and have a complex molecular phenotype [33], but the underlying genetic abnormalities are unclear and still being investigated. To date, only one study has matched the transcriptome of BTBR mice with transcriptomic data from ASD patients [34]. So far, only some features of SCZ have been modeled in the BTBR strain. Social withdrawal as a consequence of disrupted sociability is the main early symptom of both ASDs and SCZ. In this context, BTBR mice have been employed for research into the social dysfunction relevant to SCZ [35]. Reduced GABA and increased glutamate concentrations in the prefrontal cortex (PFC)—as well as weakened maturation of GABA circuits accompanied by impairments of multisensory integration—reflect the features of an SCZ-like phenotype in BTBR mice [35,36]. Moreover, one of the major genetic risk factors of SCZ, a spontaneous deletion of the *Disc1* gene, is also present in BTBR mice [37]. Because BTBR mice are often used to explore various therapeutic strategies aimed at alleviating autistic-like symptoms, it is important to understand the degree of genetic convergence between this animal model and both ASD and SCZ patients.

Here, we collate recently published data from independent transcriptomic and proteomic studies on brain samples from ASD and SCZ patients as well as the popular autistic-like animal model: BTBR mice. The main purpose of this review is to combine these datasets in one bioinformatic analysis. First, we examined these data to identify common differentially expressed genes (DEGs). Next, we explored proteomic datasets to identify differentially expressed proteins. Finally, these genes and proteins were characterized with respect to enrichment in gene–gene and protein–protein interaction networks to investigate the link with canonical pathways and biological processes implicated in the etiology of both ASDs and SCZ and shared with BTBR mice.

## 2. Materials and Methods

### 2.1. Analysis of Publicly Available Transcriptomic and Proteomic Data

To find reproducible alterations in the transcriptome of BTBR mice, we conducted literature searches on Google Scholar and PubMed (NCBI). We utilized the following search query: (BTBR); AND (cortex), OR (striatum), OR (hippocampus); AND (RNA-seq) or (microarray). This search was completed on 26 June 2022 and produced 1642 hits on Google Scholar and 661 hits on PubMed. After screening the papers, seven studies were found to meet the following criteria: (1) the experiment was performed on mice of BTBR and C57BL/6 strains; (2) tissue samples of the cortex, striatum, or hippocampus were examined; (3) a differential gene expression analysis of BTBR mice compared to C57BL/6 mice was performed; (4) proteomic or transcriptomic analysis (RNA-seq or microarray) was used (Table 1); and (5) the paper was written in the English language.

### 2.2. Functional Annotation of Reproducible Genes in BTBR Mice

With our inclusion criteria, we found three transcriptomic studies on the hippocampus and four on the cortex and/or striatum and two proteomic studies on the cortex and hippocampus (Table 1). Between the hippocampal datasets, we compared DEG sets, and changes in expression that were unidirectional between at least two datasets were designated as reproducible in BTBR mice. Because the striatum and frontal cortex are closely related structures and have many functional connections with each other and because some transcriptomic studies were performed on a tissue common between these structures, we decided to combine the data obtained from the striatum and cortex. In total, DEG sets from the cortex and/or striatum were compared between four datasets, and the criteria for reproducible changes in expression were the same as those for the hippocampus (changes in expression that were unidirectional between at least two datasets).

Gene ontology (GO) enrichment analysis of the reproducible DEGs was conducted using the enrichGO function from the Cluster-Profiler (v4.0.5) R package (v4.1.0). Basic workflow is shown in Figure 1.

### 2.3. Association of a Genetic Background and Gene Expression

We tested how genetic variants of BTBR mice affect gene expression. For this purpose, we employed BTBR mice’s sequenced genome data available in the Trust Sanger Institute mouse genome project (https://www.sanger.ac.uk/data/mouse-genomes-project/ accessed on 1 November 2022). The final list of mutations (SNPs, insertions, deletions, CNVs, and structural variants) included 785,695 rs. We analyzed the potential significance of these rs using the Ensembl Variant Effect Predictor Web interface with default parameters [44]. Next, we determined which of the genes affected by these mutations are expressed (at least 10 counts in each sample) in the mouse cortex and hippocampus. To this end, we compared genes having “high prediction score” mutations with genes from previously published datasets on C57BL/6 adult mice [45,46], which included 14,989 and 14,901 genes expressed in the hippocampus and PFC, respectively. The list of expressed genes carrying high-prediction-score mutations was analyzed for enrichment with GO terms (by means of the enrichGO function from the Cluster-Profiler (v4.0.5) R package v4.1.0) and compared with the set of reproducible DEGs.

### 2.4. Reanalysis of the Published RNA-Seq Data from the Cortex and Hippocampus of BTBR Mice

Raw data from two studies that involved RNA-seq to evaluate transcriptomic changes were reanalyzed via gene set enrichment analysis (GSEA) and via examination of alternative splicing changes. The sequencing data were preprocessed in fastp v0.20.1 [47] to remove adapters and low-quality sequences. The preprocessed data were mapped to the *Mus musculus* GRCm38 reference genome assembly in the HISAT2 aligner software, v2.2.1 [48]. The HISAT2 alignments were quantified by means of featureCounts v2.0 [49]. The quality of the sequencing data was assessed using FastQC (v0.11.9) and Picard Collec-tRnaSeqMetrics (v2.18.7) software. The aligned data having fragments per kilobase of transcript per million fragments mapped (FPKM) > 0.1 were then converted into per-gene count tables with the help of GENCODE vM22 gene annotation data. Genes were then subjected to an analysis of differential gene expression via the DESeq2 R v4.1.0 package [50]. Genes with a *p* value < 0.05 were designated as statistically significant DEGs.

GSEA was conducted using the gseGO function of the ClusterProfiler (v4.0.5) R package (v4.1.0). Genes were ranked by log2 (fold change) from DESeq2 results. In the results, a normalized enrichment score indicated whether the genes were mostly up- or downregulated in a given gene set.

### 2.5. A Comparison between BTBR-Related Genes and Genes Associated with Human Autism or SCZ

We compared data from our study with results of two meta-analyses of RNA-seq data obtained from five independent cortical datasets (postmortem tissue samples) of ASD patients and two meta-analyses of microarray data obtained from eight independent cortical datasets of SCZ patients (Table 2). Gene orthologs were identified by means of BioMart (https://www.ensembl.org/biomart/martview/ accessed on 1 November 2022). Genes that were differentially expressed (*p* < 0.05) unidirectionally between two meta-analyses were regarded as associated with either ASD or SCZ.

## 3. Results

### 3.1. Transcriptome Aberrations in the Hippocampus of BTBR Mice

The comparison of sets of up- and downregulated genes from various authors revealed 56 upregulated genes and 135 downregulated genes whose expression changed in at least two of the three studies included in the analysis (Appendix A). In this gene set, we detected one enriched GO term at a false discovery rate (FDR) of <0.05: “oxidoreductase activity” (related to genes *Adi1*, *Ptgs2*, *Alox8*, and *Alox12b*). Expression of 4 upregulated genes (*Blvrb*, *Scg5*, *Serpina3n*, and *Anxa5*) and 14 downregulated genes (*C1qb*, *Spink8*, *Entpd4*, *Pop4*, *Alg1*, *Rpl29*, *Ccnd1*, *Mt3*, *Zfp131*, *6330403K07Rik*, *Nudt19*, *C1ql2*, *Evc2*, and *Cetn4*) changed unidirectionally among all three datasets (Figure 2a).

Among these genes, there are genes encoding complement components (*C1qb* and *C1ql2*), inhibitors of serine peptidases (*Serpina3n* and *Spink8*), and a neuron-specific gene (*Scg5* encoding neuroendocrine protein 7B2 involved in the regulation of the corticotropin secretory pathway). Consistent changes in the expression of these genes in a number of studies indicated that they constitute a molecular signature of the hippocampal transcriptome of BTBR mice. Nonetheless, functions of the protein products of these genes are diverse because they are not components of a specific pathway(s).

A comparison of the transcriptome data with the proteome data showed that only two genes undergo unidirectional expression changes at both mRNA and protein levels. One of them is the Wfs1 gene—whose protein product participates in protein biosynthesis, stabilization, folding, maturation, and secretion [55]—and the other gene is Clcn6, which codes for a chloride/proton exchanger playing an important role in autophagic-lysosomal function [56]. Of note, mitochondrial phosphate carrier (Slc25a3) mRNA was underexpressed while the protein was overexpressed. 

Next, to identify the functional pathways or signatures in the set of all expressed genes, we performed GSEA and found that in the hippocampus of BTBR mice, there was overexpression of genes associated with GO terms (biological processes) “sensory perception of smell” and “olfactory receptor activity” and underexpression of genes related to regulation of “corticotropin secretion” (Figure 2e, Appendix A).

### 3.2. Transcriptome Alterations in the Corticostriatal Area of BTBR Mice

We identified 79 upregulated genes and 113 downregulated genes whose expressions were found to change in at least two of the four studies included in the analysis (Appendix A). In this gene set, there were two enriched GO terms with FDR < 0.05: “actin cytoskeleton” (related to such genes as *Crocc*, *Myh6*, *Myh7*, and *Myo7a*) and “humoral immune response” (related to genes *C1qb*, *C1qc*, *B2m*, *Masp2*, *Cci17*, *Slpi*, *Ptprc*, and *Pglyrp1*; Figure 3c). The expression of six genes changed unidirectionally among all datasets (Figure 3a); among these genes, an increase in expression (*Scg5* and *Serpina3n*) and a decrease in expression (*C1qb*, *Pop4*, and *Nudt19)* were also found in all hippocampal datasets, suggesting that these changes are consistent across different experimental designs and not specific to one brain structure. The expression of *Anxa3*, which encodes a calcium-dependent phospholipid-binding protein involved in signal transduction pathways, was low only in the corticostriatal region.

To understand the similarity of the molecular signatures between the hippocampal transcriptome and corticostriatal transcriptome, we compared consistent changes in the expression of genes (a change in expression unidirectional between at least two datasets) and noticed that the expression of 55 genes (13 upregulated and 42 downregulated) changed in the same way between these brain structures (Figure 4). At the same time, a comparison of proteomic changes between the cortex and hippocampus suggested that protein expression of only four genes (*Macf1*, *Bsn*, *Psd3*, and *Chchd3*) changed in the same direction between these brain regions. Of note, a comparison of the transcriptome data with the proteome data did not uncover expression alterations that were unidirectional between mRNA and protein levels in either the cortex or striatum.

Next, to identify functional pathways or signatures in the set of all expressed genes, we performed GSEA and noticed that in the striatum of BTBR mice, there was upregulation of genes associated with the GO terms (biological processes) “motile cilium” and “microtube bundle formation” and downregulation of genes related to “innate immune response” (Figure 3e, Appendix A).

### 3.3. Genetic Characteristics of BTBR Mice

We found 378 mutations that had a high prediction score (Appendix A). This list included 261 mutations that lead to a frameshift, 108 mutations that affect splice donor sites, and four mutations located in the 3′ untranslated region and resulting in the loss of a stop codon. Among the genes expressed in the hippocampus and cortex, only 124 and 120 genes had at least one mutation with a high-prediction-score predictor impact. These sets of genes turned out to be not enriched with any GO terms. Unexpectedly, we found only six DEGs in the hippocampus and seven DEGs in the corticostriatal area that were also in the set of expressed genes containing high-prediction-score mutations (Figure 1b and Figure 2b). Among these genes, the expression of *Alg1*, *Tmem260*, and *Zmynd11* was low, while the expression of *Abhd1* and *Olfml1* was high in both brain structures. In these genes, the presence of high-prediction-score mutations—which can lead to a frameshift or affect a splice site—can result in a defective protein product. Thus, regardless of the level of expression of these genes, the functionality of the encoded protein products may be impaired.

### 3.4. The Comparison between DEGs from Postmortem ASD or SCZ Human Tissue Samples and DEGs from BTBR Mice

After comparing the results of meta-analyses of postmortem samples from patients with SCZ or autism, we identified 226 ASD-related genes and 155 SCZ-related genes in the frontal cortex (Appendix A). The comparison of these gene sets with the reproducible DEGs found in the corticostriatal area of BTBR mice yielded only a small overlap (Figure 5). Only the expression of *Lpl* proved to be upregulated in postmortem samples from SCZ patients and in BTBR mice. *Lpl* codes for an enzyme called lipoprotein lipase, which plays a key part in the brain energy balance. 

The set of DEGs from BTBR mice shared four genes with the set of ASD-related genes (*Gpr84*, *Oscar*, *Ptprc*, and *Eps8l1*), but their expression alterations were not unidirectional (upregulated in ASD and downregulated in BTBR mice). *Gpr84* and *Ptprc* are microglia-specific genes [57] and encode receptors involved in the regulation of neuroinflammation. *Ptprc* encodes common lymphocyte antigen CD45 playing an important role in T cell activation [58]. *Gpr84* participates in the modulation of the inflammatory response, and upregulation of its expression has been documented during a response to inflammatory conditions and stimuli [59].

## 4. Discussion

Our review includes transcriptomic and proteomic data from the hippocampus and corticostriatal area. Functional and neuroanatomical changes in these brain structures are most often reported in patients with an ASD. Much evidence points to atypical cortical morphology, volume [60,61], and grey and white matter thickness [62,63,64,65] as well as microstructure abnormalities [66] in individuals with an ASD. A number of studies indicate abnormal enlargement [67,68,69,70,71,72,73,74], shape asymmetry [75], and altered functional connectivity [76] in the hippocampus of ASD subjects. Similar alterations of shapes [77,78], volumes [79,80,81,82,83], microstructure [84], and connectivity [85,86] have been found in basal ganglia, especially in striatal subregions.

Neuroanatomical and functional changes of the cortex [87,88,89,90], hippocampus [91,92], and striatum [93,94,95] are also present in animal models of ASDs but can vary from one model to another [96]. Neuroanatomic features of BTBR mice as a model of idiopathic ASDs are relatively well characterized. Apart from their most striking feature—the absence of the corpus callosum [97]—BTBR mice also have a deficient dorsal hippocampal commissure, smaller hippocampal volume [98,99], and dramatically reduced thickness and volume of the cortex, particularly in the PFC [99]. Smaller gray matter volume in various cortical and subcortical areas has been reported in MRI studies [98,100,101]. It is worth noting structural [100] and functional [102,103] deviations in the striatum of BTBR mice. Marked changes in cortical and subcortical connectivity in BTBR mice have been registered as well [89,102,104]. Overall, the aforementioned impairments not only are consistent with the behavioral phenotype of BTBR mice but also recapitulate neuroimaging hallmarks of autism. Considerable changes in neuroplasticity inevitably underlie these neuroanatomical abnormalities.

We found that the transcription of genes *Scg5* and *Serpina3n* is overactive and that the transcription of *C1qb*, *Pop4*, and *Nudt19* is low in all datasets from the hippocampus, cortex, and striatum of BTBR mice. This finding indicates that *Scg5*, *Serpina3n*, *C1qb*, *Pop4*, and *Nudt19* are universal transcriptional markers of BTBR mice. 

The 7B2 protein, the product of the *Scg5* gene, is known as a secreted chaperone whose expression is restricted to neurons and neuroendocrine and endocrine cells [105,106]. 7B2 not only is crucial for peptide hormone storage [107,108] but also has antiaggregant properties and is capable of reducing the fibrillation of aggregating proteins [109,110]. Some contradictory pieces of evidence are suggestive of aberrant 7B2 levels in patients with Alzheimer’s disease [110,111,112]. Furthermore, Helwig et al. [110] have documented colocalization of 7B2 with α-synuclein deposits in brain samples from patients with Parkinson’s disease. Currently, there is no proof of the involvement of 7B2 in the pathogeneses of neurodevelopmental disorders. In one study on autistic subjects in the Japanese population, researchers did not find a correlation between a deletion in chromosomal region 15q11–q13 (containing *SCG5*) and autism [113]. 

In the present work, we noticed unidirectional changes between *Wfs1* transcription and protein levels. WFS1 is a membrane protein that is vital for the transfer of vesicular cargo proteins from the endoplasmic reticulum to the Golgi apparatus and is associated with diabetes [114]. There is evidence that *Wfs1* regulates proper folding of 7B2 [115]. Although the link between *Wfs1* and autism is still unproven, it has been demonstrated that a mutation in *WFS1* causes Wolfram syndrome, which is associated with bipolar disorder and SCZ [116,117,118]. Wolfram syndrome itself is characterized by various neurological problems, including ataxia, seizures, hypersomnolence, brain stem atrophy, peripheral and autonomic neuropathy, and an inability or decreased ability to sense taste and odors [119]. Recently, it was shown that WFS1 directly interacts with SCG5 vesicular cargo protein in pancreatic β-cells. In the brain, they potentially could be involved in the process of sorting neuropeptide cargo proteins into the COPII vesicles. However, this is yet to be proved [114]. Collectively, these data imply that *Scg5* and *Wfs1* are relevant to proteostatic processes, and it is possible that in the neuronal system of BTBR mice, proteostasis is compromised.

*Serpina3n* encodes the SerpinA3N protein belonging to the serpin superfamily of protease inhibitors. Upregulation of transcription of *Serpina3n* is a strong marker of reactive astrogliosis [120]. Some research articles show anti-inflammatory and neuroprotective effects of SerpinA3N [121,122,123], whereas in one study, the opposite was demonstrated [124]. SERPINA3 upregulation has been detected in postmortem samples of the cortex from SCZ patients [125]. It is not clear whether high levels of SerpinA3N can also accompany other neurodevelopmental disorders, but we can hypothesize that elevated transcription of *Serpina3n* in different brain structures of BTBR mice is an indicator of neuroinflammatory processes.

Our findings suggest that the complement and coagulation cascade signaling pathway are also affected in the brain of BTBR mice because the transcription of *C1qb* proved to be low in all the analyzed brain structures. Together with other components of the classical complement cascade (C1qa, C1qc, C2, and C4), C1qb is expressed by uninjured peripheral nerves and is known to play a crucial part in myelin clearance after peripheral nerve injury [126]. Reactivation of compliment expression can induce or propagate inflammation and may be detrimental to peripheral nerves [127]. *C1qb* is also implicated in the proinflammatory response in the central nervous system [128,129,130,131]. It is of note that in DBA/2J mice, which exhibit a deficit of social interaction and are known as a model of SCZ-related behavior, *C1qb* transcription is lower in the cortex, hippocampus, and hypothalamus as compared to C57BL/6N mice [132]. Consequently, changes in the transcription of *C1qb* in BTBR mice also support the involvement of inflammation in the pathogenesis of their autistic-like phenotype. These results are also consistent with previous studies in which BTBR mice have been shown to have an impaired immune response [133,134]. 

Pop4 is a subunit of a ribonucleoprotein enzyme called ribonuclease P (RNase P): an essential enzyme that catalyzes the removal of the 5′ leader sequence from precursor tRNAs [135]. There is lack of information on the role of Pop4 in neuronal functions and neuropathology. In general, the evidence of RNase P participation in neuronal function is scarce [136,137,138,139]. In a paper by Cai et al. [137], it is demonstrated that a knockdown of *Rpph1* diminishes dendritic spine density in primary culture of hippocampal pyramidal neurons. One report suggests that in the PFC of autism patients, transcription of *RPP25* is low [140]. The weak *Pop4* transcription in the brain of BTBR mice also suggests that RNase P may be implicated in the development of the autistic-like phenotype in these animals. 

Nudt19 is a member of the Nudix hydrolase family with an RNA-decapping activity [141,142]. Additionally, Nudt19 activity controls coenzyme A degradation [143,144] and fatty acid oxidation in hepatic cells [145]. In the hepatocellular carcinoma cell line, Nudt19 activated the mTORC1–P70S6K signaling pathway [146]. To date, there have been only two studies on Nudt19 in the context of neuronal functions. On the basis of whole-genome microarray analyses, Arisi et al. [147] proposed that Nudt19 is a potential biomarker of the early stage of Alzheimer’s-disease-like neurodegeneration in mice. Recently, an analysis of the striatal proteome of depression-susceptible and anxiety-susceptible and -insusceptible rat cohorts detected Nudt19 among abnormally expressed proteins [148]. Although the exact functions of Nudt19 inside neuronal cells are still unclear, the regulation of cellular bioenergetics and of mTOR signaling by Nudt19 is an intriguing topic and implies a major role in neuronal functions and neuropathology as well.

Aside from *Wfs1*, our comparison of transcriptomic and proteomic data revealed unidirectional expression changes for the *Clcn6* gene in the hippocampus. Mutations in *Clcn6* in mice lead to mild lysosomal storage abnormalities, whereas in humans, a *CLCN6* mutation causes a severe developmental delay with pronounced neurological and neurodegenerative problems [149]. Proper functioning of Cl^−^/H^+^ exchangers is important for adequate activities of endosomes and lysosomes. Even in the presence of mild lysosomal abnormalities, in Clcn6^−/−^ mice, abnormal storage of some substances causes pathological enlargement of proximal axons [150]. The marked astrocytosis in the cortex and lowered corpus callosum volume are also seen in Clcn6^−/−^ mice [151]. 

Our proteomic data indicate that changes in the protein expression of *Macf1*, *Bsn*, *Psd3*, and *Chchd3* are unidirectional between the cortex and hippocampus. The *Chchd3* gene encodes a mitochondrial protein located in the intermembrane space and essential for maintaining crista integrity and mitochondrial function [152]. One article points to an association between CNVs in *CHCHD3* and an ASD [153]. 

Protein products of genes *Macf1*, *Bsn*, and *Psd3* are associated with cellular morphology and synaptic function. Microtubule–actin crosslinking factor 1 (MACF1) is a cytoskeleton-crosslinking protein that interacts with microtubules and F-actin to modulate the polarization of cells and coordination of cellular movements [154]. Furthermore, MACF1 participates in the Wnt–β-catenin signaling pathway. It is not surprising that MACF1 is important for cell migration, which requires continuous reconstruction of the cytoskeleton [155,156]. A number of reports indicate a contribution of MACF1 mutations to different neurological disorders including SCZ [154]. In the context of an ASD as a neurodevelopmental disorder involving aberrant neuronal migration [157], it is highly likely that MACF1 deregulation may also participate in the pathological changes. Pleckstrin and sec7 domain-containing 3 (PSD3), also known as EFA6R, regulates localization of small GTPase ARF6, thereby promoting a cytoskeletal rearrangement [158]. At least in humans, PSD3 localization is restricted to the PFC [159], and *PSD3* has been identified among candidate genes related to Alzheimer’s disease pathophysiology [160]. PSD3 was identified as one candidate of ASD-associated genes in a duplicated locus of chromosomal region 8p22-21.3 in ASD patients [161]. 

Bassoon (Bsn) is the presynaptically localized scaffolding protein that is a negative regulator of presynaptic autophagy and of the ubiquitin–proteasome system in the presynapse [162]. Research on bassoon knockout mice and cultured neurons indicates that this protein is a key regulator of synaptic vesicle proteostasis [163,164]. A conditional knockout of *Bsn* exclusively in forebrain excitatory neurons enhances hippocampal excitability and neurogenesis [165], which may be TrkB-dependent [166]. Participation of bassoon in a number of neuropathologies has been demonstrated. Inflammation-induced accumulation of bassoon in the central nervous systems of mice and humans boosts neurotoxic processes in multiple sclerosis [167]. A rare mutation in *BSN* correlates with a familial type of SCZ [168]. 

The abovementioned reproducible genes identified in transcriptomic and proteomic datasets fit well the functional pathways revealed by GSEA. For example, “sensory perception of smell” and “olfactory receptor activity” are consistent with the involvement of *Wfs1* in olfaction [169,170,171]. This finding is in good agreement with neuroanatomical anomalies in olfactory bulbs [98,100] and low capacity for odor discrimination in BTBR mice [172,173,174]. Likewise, the underexpression of genes associated with the regulation of corticotropin secretion is in agreement with corticosteroid dysregulation [175,176] and excessive stress hormone responses in BTBR mice [177,178]. The protein products of genes *Wfs1* and *Scg5* may directly take part in the above-mentioned pathways. Similarly, changes in the expression of *Macf1*, *Bsn*, and *Psd3* are consistent with “motile cilium” and “microtube bundle formation” pathways, and the *C1qb* transcription change is concordant with “innate immune response” uncovered by GSEA. Again, in BTBR mice, we can see relevant phenotypic changes linked with neuronal morphology and migration [97,179,180] and with the immune system [181,182,183]. 

Our comparison of the results of meta-analyses of DEGs in postmortem samples from SCZ or autism patients with reproducible DEGs in the corticostriatal area of BTBR mice yielded only a small overlap. Despite the substantial overlap between ASD and SCZ genetic risk factors in humans [18,28], such convergence is absent in mice. Nonetheless, it should be taken into account that the comparison of results on gene and protein expression between BTBR mice and postmortem samples from individuals with SCZ or autism has a number of limitations in terms of data interpretation. The first set of limitations has to do with the heterogeneity of experimental data both in humans (age, severity of the disease, genetics, and comorbidities) and in mice (age, coordinates of brain structures, sample preparation, and various data analysis algorithms). Postmortem data also suffer from small sample sizes and lack of ethnic diversity because they are composed primarily of subjects with European or North American genetic backgrounds [184]. Differences in the design of experiments mean that we can analyze only the most pronounced aberrations. The second limitation is brain morphofunctional differences between rodents and humans. These dissimilarities are observed both in adulthood and in the neonatal period, when the stage of mouse brain development corresponds to the stage of development in the third trimester of pregnancy in humans [185]. Presently, it is not possible to unambiguously match different regions of the cerebral cortex between rodents and humans. On the other hand, the “multi-omics” approach employed here to characterize genetic architecture of BTBR mice uncovered some convergence between genes and pathways implicated in ASDs and in the autistic-like behavior of BTBR mice. Mahony and O’Ryan [23], in their review of proteomic, transcriptomic, and epigenomic data from postmortem brain samples from ASD patients, identified four canonical pathways enriched within seven or more independent datasets. mTOR signaling, oxidative phosphorylation, adipogenesis, and complement response pathways, which were revealed by Mahony and O’Ryan, are relevant to some genetic hallmarks of BTBR mice. For example, the observed alterations of transcription of *Serpina3n*, *C1qb*, and *Nudt19* and of protein expression of *Chchd3* all point to the involvement of mTOR signaling, of the complement response, and of mitochondrial function in the pathogenesis. These expression changes in genes are likely different from those in ASD patients but lead to similar phenotypic manifestations in BTBR mice. It can be hypothesized that we have an example of convergent development of an ASD-like phenotype and an ASD phenotype.

## 5. Conclusions

In this review, for the first time, we identified reproducible alterations of gene and protein expression in different regions of the brains of BTBR mice. Our findings indicate that these molecular signatures do not reproduce the expression changes observed in postmortem samples from ASD and SCZ patients. In addition, these changes in expression do not correlate well with the genetic background of these animals. To expand our knowledge about the molecular signatures of BTBR mice, transcriptomic and epigenomic studies on individual cell populations in different brain regions are required. These data will allow us to assess the contribution of epigenetic features to the magnitude of gene expression as well as cell-specific anomalies.

## Figures and Tables

**Figure 1 biomedicines-11-00289-f001:**
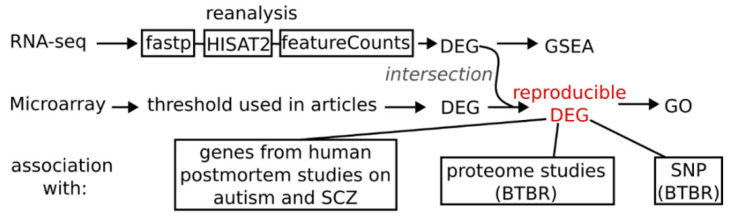
Basic workflow scheme. These steps were applied separately for hippocampus and cortex + striatum datasets resulting in two reproducible differentially expressed genes lists. For reanalysis of available RNA-seq data first adapters and low-quality reads were trimmed using fastp program, then aligned to mm10 genome with HISAT2 (2.2.1) software and summarized to gene counts with featureCounts (v2.0) function using genecode vM22 annotation.

**Figure 2 biomedicines-11-00289-f002:**
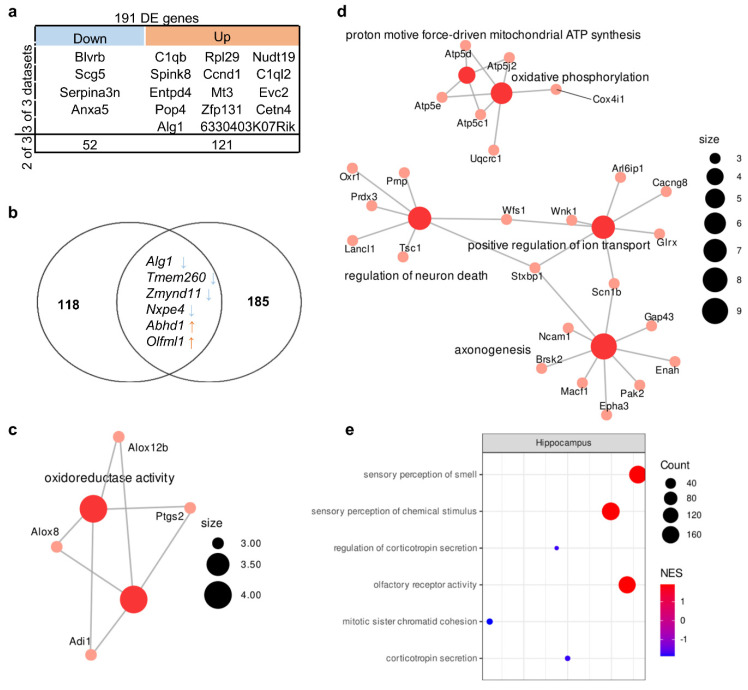
Molecular signatures in the hippocampus of BTBR mice. (**a**) Reproducible up- and downregulated genes in the hippocampus. Out of a total of 191 DE genes, 4 were downregulated in all 3 datasets, and 52 were downregulated in 2 out of 3; 14 were upregulated in all 3 datasets, and 121 were upregulated in 2 out of 3. (**b**) The overlap between the set of genes carrying a “high prediction score” mutation and the set of DEGs. Out of 124 “high prediction score” genes only 6 overlapped with reproducible DE genes set (191 gene). (**c**) The GO category enriched in the set of DEGs. Size of the node represents number of genes in the category. (**d**) The GO categories enriched in the set of differentially expressed proteins according to data from ref. [40]. Size of the node represents number of genes in the category. (**e**) Top three up- and downregulated GSEA categories in the set of all expressed genes according to reanalysis of PRJNA533538 data. NES: normalized enrichment score. Size of the circle (count) represents number of genes in the category, color of the circle represents normalized enrichment score (*NES*).

**Figure 3 biomedicines-11-00289-f003:**
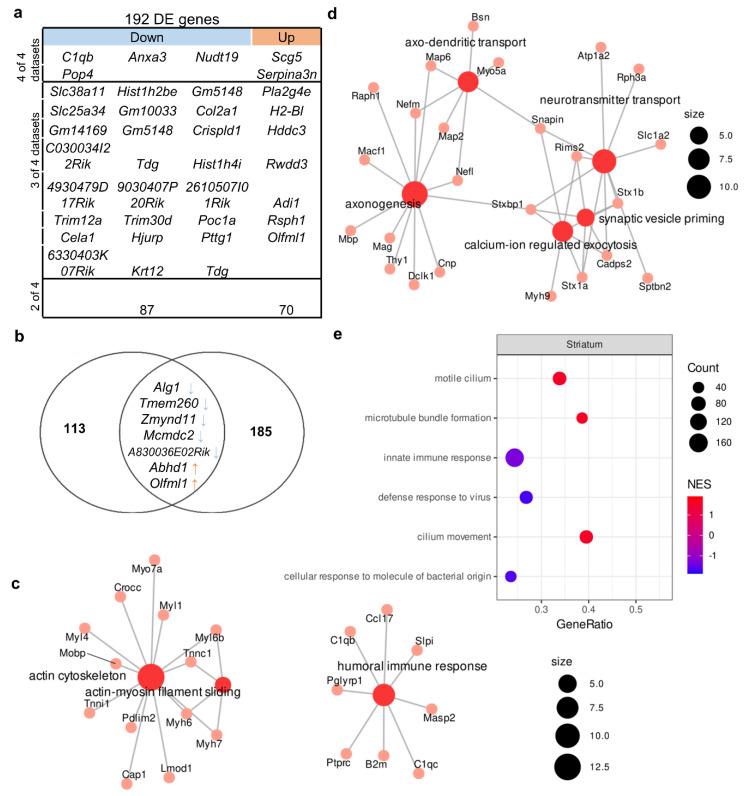
Molecular signatures in the corticostriatal area of BTBR mice. Out of a total of 192 DE genes, 4 were downregulated in all 4 datasets, and 24 were downregulated in 3 out of 4 and 87 in 2 out of 3. In all datasets, 2 genes were upregulated, 7 genes were upregulated in 3 out of 4 datasets, and 70 were upregulated in 2 out of 4. (**a**) The comparison of DEGs from the corticostriatal area. (**b**) The overlap between the set of genes carrying a high-prediction-score mutation and the set of DEGs. Out of 120 “high prediction score” genes, only 7 overlapped with reproducible DE genes set (192 gene). (**c**) The GO category enriched in the set of DEGs. Size of the node represents number of genes in the category. (**d**) GO categories enriched in the set of differentially expressed proteins on the basis of data from Ref. [43]. Size of the node represents number of genes in the category. (**e**) Top three up- and downregulated GSEA categories in the set of all expressed genes according to the reanalysis of GSE138539 data. NES: normalized enrichment score. Size of the circle (count) represents number of genes in the category, color of the circle represents normalized enrichment score (NES).

**Figure 4 biomedicines-11-00289-f004:**
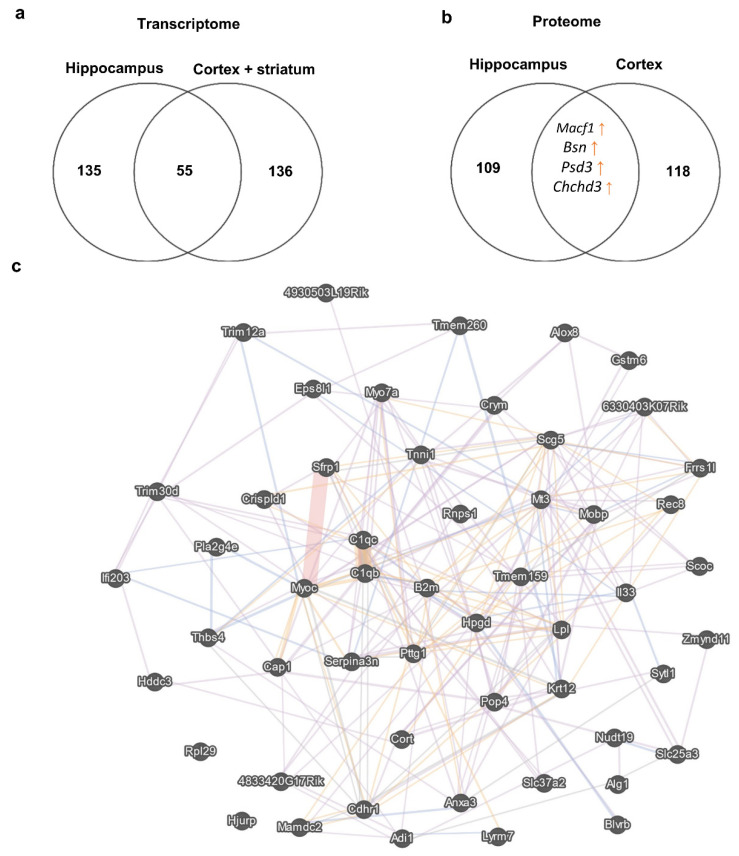
DEGs in the corticostriatal area and hippocampus. (**a**) This Venn diagram depicts the overlap between sets of reproducible corticostriatal and hippocampal genes changing their expression unidirectionally between these brain regions. Numbers on the diagram represent how many genes are unique for the dataset and how many overlap. (**b**) This Venn diagram illustrates the overlap between sets of cortical and hippocampal proteins changing their expression unidirectionally between these brain regions. Numbers on the diagram represent how many genes are unique for the dataset and how many overlap. (**c**) Gene mania network representation of the 55 reproducible corticostriatal and hippocampal genes changing their expression unidirectionally between these brain regions; each gene manifested expression changes in at least two datasets from the hippocampus and corticostriatal area.

**Figure 5 biomedicines-11-00289-f005:**
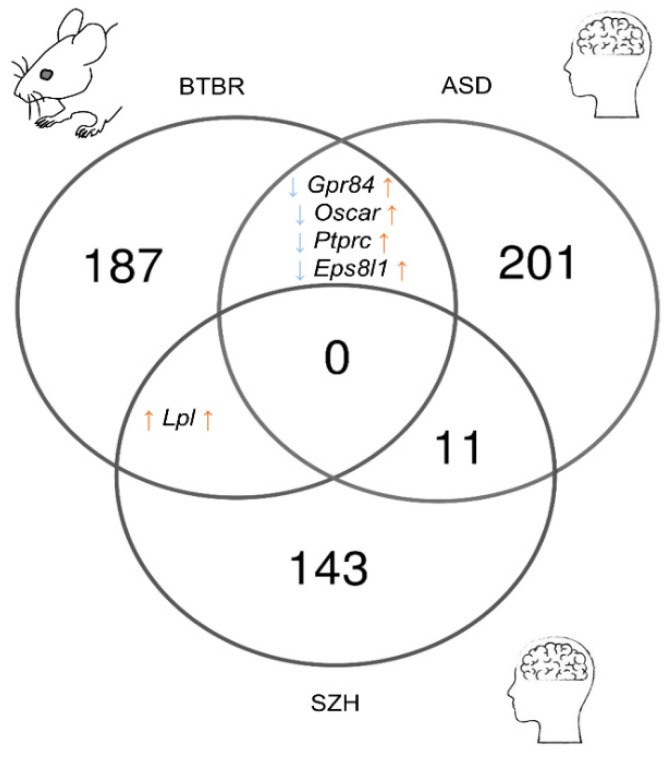
The comparison of reproducible DEGs between the corticostriatal area of BTBR mice and postmortem samples from the frontal cortex of SCZ and ASD patients. Numbers on the diagram represent how many genes are unique for the dataset and how many overlap.

**Table 1 biomedicines-11-00289-t001:** The transcriptomic and proteomic studies on BTBR mice included in the analysis.

Strains	Age	Brain Region	Method	Threshold	Raw Data	Reference
BTBR T + Itpr3tf/J (BTBR)/C57BL/6J	12 w	Hippocampus	RNA-seq	*p* value < 0.05 *	PRJNA533538	[38]
BTBR T+ Itpr3tf/J (BTBR)/C57BL/6J	3–5 m	Hippocampus	Microarray	Rank Product (RP) non-parametric method was used	GSE81501	[39]
BTBR T + Itprtf/J mice/C57BL/6J	4 m	Hippocampus and cortex	Microarray	Z-ratio value of ±1.50 and/or a Z-test value *p* < 0.05	N/A	[40]
BTBR/B6	8 w	Cortex	RNA-seq	|log_2_FC| ≥ 1 and p_adj_ ≤ 0.05	N/A	[41]
BTBR T + tf/J/C57BL/6J	8–10 w	Striatum (bregma −0.58–1.53).	RNA-seq	*p* value < 0.05 *	GSE138539	[42]
BTBRTF/ArtRbrc mice Compared to C57BL/6J Mice	7 w	Striatum + Cortex	Microarray	p_adj_ ≤ 0.05	GSE156646	[34]
BTBR T + Itprtf/J mice/C57BL/6J	4 m	Hippocampus	iTRAQ LC–MS/MS	fold change > 0.2	N/A	[40]
BTBR T + Itprtf/J mice/C57BL/6J	8 w	Cortex	iTRAQ LC–MS/MS	fold change > 0.5, *p* < 0.05	N/A	[43]

Threshold column shows the significance cut off used for DEG assessment. In the column Raw Data can be found GEO project IDs if available or N/A (not available) if not. * RNA-seq studies with available raw data were reanalyzed for a *p* value cutoff of <0.05; in other papers, the cutoff of the original study was accepted; w: weeks, m: months.

**Table 2 biomedicines-11-00289-t002:** Publicly available meta-analyses of postmortem cortical transcriptomes of SCZ and ASD patients.

Disease	Type of Review	Data ID	Method	Threshold	Reference
ASD	Meta-analysis	GSE28475	Microarray/RNA-seq	*p* < 0.05	[51]
		GSE28521			
		GSE36192			
ASD	Meta-analysis	GSE64018	RNA-seq	*p* < 0.05	[52]
		GSE30573			
SCZ	Meta-analysis	GSE17612	Microarray	*p* < 0.05	[53]
		GSE21935			
		GSE25673			
		GSE12649			
		GSE21338			
SCZ	Meta-analysis	GSE17612	Microarray	q-value < 0.05	[54]
		GSE21138			
		+and four others			

Threshold column shows the significance cut off used for DEG assessment. DATA ID shows GEO accession numbers of experiments used in the study. ASD—autism spectrum disorder; SCZ—schizophrenia.

## Data Availability

Data supporting the findings of this study are available within the article and Appendix A here.

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
