# Peer review of "Molecular Abnormalities in BTBR Mice and Their Relevance to Schizophrenia and Autism Spectrum Disorders: An Overview of Transcriptomic and Proteomic Studies"

_biomedicines, 2023, doi:10.3390/biomedicines11020289_

Round 1

Reviewer 1 Report

This is a well written review by Kisaretova et al. summarizing the potential utility of BTBR mice models to study the molecular factors underlying schizophrenia and Autism. I have a few comments. 

1. As presented, the manuscript appears to be a hybrid between review and research papers. Considering the authors have already performed bioinformatic analyses, it will be beneficial to transform this manuscript into a full fledged research paper. 

2. Following my first comment, I suggest that the authors perform a meta-analysis of the available transcriptomic (and, if possible) on the few proteomic studies in BTBR mouse models similar to what they have done on the human postmortem brain studies. With the increased power, they might be able to identify additional loci that the individual studies could have missed. 

3. In the methods section 4.3, the authors describe their approach to testing the association between genetic variants and their impact on gene expression. Ideally, this is usually conducted via eQTL, or pQTL analyses, yet no such results were reported. While using the Ensembl variant effect predictor to identify variants with significant impact on protein function is interesting, by now, it is well accepted that virtually almost all common polymorphisms identified by various GWAS in Schizophrenia and Autism have far more subtle effects on gene function.

4. Having access to the raw gene expression data, the study would have benefited from performing additional network analyses such as WGCNA. Furthermore, within WGCNA they could have performed network preservation analyses to identify unique or shared gene networks between the different brain regions or between the human and mouse data, if possible. 

5. Some additional minor comments include the parameters of the variant effect predictors and criteria for high prediction scores. On lines 55-56, I would replace "core" genes with just genes and "peripheral" genes with gene modifiers.

Author Response

Dear Referee:

We are thankful to you for reviewing manuscript biomedicines-2140311 and sharing your valuable comments and concerns with us.

According to the comments, we replaced figures in bad resolution with higher quality figures, changed section order in accordance with paper guidelines, added workflow picture, and elaborated on genes involved in ASD and SCZ topic. Our point-by-point responses to the Reviewers’ comments are presented below.

We believe that the manuscript in its present form is substantially better.

Sincerely,

Vasiliy Reshetnikov

  1. As presented, the manuscript appears to be a hybrid between review and research papers. Considering the authors have already performed bioinformatic analyses, it will be beneficial to transform this manuscript into a full fledged research paper.
  2. Following my first comment, I suggest that the authors perform a meta-analysis of the available transcriptomic (and, if possible) on the few proteomic studies in BTBR mouse models similar to what they have done on the human postmortem brain studies. With the increased power, they might be able to identify additional loci that the individual studies could have missed.

Reply: Thank you for this comment. Initially we planned to do reanalysis for both RNA-seq and microarray studies and then integrate them in meta-analysis. But we couldn’t get a hold on raw data or full gene lists with fold changes and p values of some studies. Therefore, unfortunately this is impossible at the moment because we have only 2 RNA-seq datasets available, each made on different murine brain structures.

  1. In the methods section 4.3, the authors describe their approach to testing the association between genetic variants and their impact on gene expression. Ideally, this is usually conducted via eQTL, or pQTL analyses, yet no such results were reported. While using the Ensembl variant effect predictor to identify variants with significant impact on protein function is interesting, by now, it is well accepted that virtually almost all common polymorphisms identified by various GWAS in Schizophrenia and Autism have far more subtle effects on gene function.

 Reply: Thank you for the suggestion. We looked into it and learned that in order to perform eQTL analysis we need transcriptomic data from more than 2 mouse strains, because 2 data points are not sufficient for logistic regression that is used in eQTL. So, it is necessary to extend sample size by adding other strains. Though it would be interesting to perform such analysis in the future on the greater number of different mouse strains, probably as a separate study.

  1. Having access to the raw gene expression data, the study would have benefited from performing additional network analyses such as WGCNA. Furthermore, within WGCNA they could have performed network preservation analyses to identify unique or shared gene networks between the different brain regions or between the human and mouse data, if possible.

Reply: Indeed, such analysis would be beneficial if we had several transcriptomic studies or raw microarray data. Unfortunately, right now we only have access to one RNA-seq study of murine striatum and one hippocampus and we think that WGCNA performed on such small sample number wouldn’t be very informative.

  1. Some additional minor comments include the parameters of the variant effect predictors and criteria for high prediction scores. On lines 55-56, I would replace "core" genes with just genes and "peripheral" genes with gene modifiers.

Thank you for suggestion, we have inserted the requested information into the manuscript. Concerning the retrievement of CNVs, we used Ensembl Variant Effect Predictor Web interface with default parameters. We added this information in the methods section.

Reviewer 2 Report

My suggestions:

1. I would write in the introduction a few examples of genes, involved in ASD and SCZ. It would be nice if the authors would mention some examples, which may affect both SCZ and ASD. 

2. In Methods in Table 1 (or in a separate table), I would add a few examples of variants and differentially expressed genes, described in each study.

3. Figure 1d may be uploaded in better resolution. 

4. In the Methods section I would add a workflow figure. 

5. I would introduce the Wfs1 functions more in detail in the discussion. Also, I would add a brief explanation of the common pathways between Wfs1 and Scg5.

Author Response

Dear Referee:

We are thankful to you for reviewing manuscript biomedicines-2140311 and sharing your valuable comments and concerns with us.

According to the comments, we replaced figures in bad resolution with higher quality figures, changed section order in accordance with paper guidelines, added workflow picture, and elaborated on genes involved in ASD and SCZ topic. Our point-by-point responses to the Reviewers’ comments are presented below.

We believe that the manuscript in its present form is substantially better.

Sincerely,

Vasiliy Reshetnikov

#2

My suggestions:

  1. I would write in the introduction a few examples of genes, involved in ASD and SCZ. It would be nice if the authors would mention some examples, which may affect both SCZ and ASD.

Reply: We have added this information into the text.

  1. In Methods in Table 1 (or in a separate table), I would add a few examples of variants and differentially expressed genes, described in each study.

Reply: Thank you for the suggestion. The goal of our study was to find most reproducible genes in neurodevelopmental disorders and we think that showing divergent genes can potentially set the wrong focus. Description of these genes could be found in original papers linked in Table1. Also we don’t want to repeat this information there in order not to bulk up the table.

  1. Figure 1d may be uploaded in better resolution.

Reply: Thank you for noticing. We replaced all low resolution figures with better ones.

  1. In the Methods section I would add a workflow figure.

Reply: Workflow figure was added in the Methods section.

  1. I would introduce the Wfs1 functions more in detail in the discussion. Also, I would add a brief explanation of the common pathways between Wfs1 and Scg5.

This description is now provided (page 11, line 342-352).

Round 2

Reviewer 1 Report

The authors have answered all my concerns; I do not have any additional comments

Author Response

Dear Referee:
We are thankful to you for reviewing manuscript biomedicines-2140311 and sharing your valuable comments and concerns with us.
According to the comments we added legends to figures and clarified non-obvious parts.

Sincerely,
Vasiliy Reshetnikov

Reviewer 2 Report

The manuscript is acceptable. 

Author Response

(The authors gave the same response as above.)
